# The lived experience of chronic headache: a systematic review and synthesis of the qualitative literature

Vivien P Nichols,[1] David R Ellard,[1] Frances E Griffiths,[2] Atiya Kamal,[1] Martin Underwood,[1] Stephanie J C Taylor,[3] on behalf of the CHESS team

[1]Warwick Clinical Trials Unit, Division of Health Sciences, Warwick Medical School, University of Warwick, Coventry, UK
[2]Division of Health Sciences, Warwick Medical School, University of Warwick, Coventry, UK
[3]Complex Intervention and Social Practice in Health Care Unit, Centre for Primary Care and Public Health, Blizard Institute, Barts and The London School of Medicine and Dentistry, Queen Mary University of London, Coventry, UK

**Correspondence to**
Vivien P Nichols;
v.p.nichols@warwick.ac.uk

## ABSTRACT

**Objective** To systematically review the qualitative literature of the lived experience of people with a chronic headache disorder.

**Background** Chronic headaches affect 3%–4% of the population. The most common chronic headache disorders are chronic migraine, chronic tension-type headache and medication overuse headache. We present a systematic review and meta-ethnographic synthesis of the lived experience of people with chronic headache.

**Methods** We searched seven electronic databases, hand-searched nine journals and used a modified Critical Appraisal Skills Programme checklist to appraise study quality. Following thematic analysis we synthesised the data using a meta-ethnographic approach.

**Results** We identified 3586 unique citations; full texts were examined for 86 studies and 4 were included in the review. Included studies differed in their foci: exploring, patient-centred outcomes, chronic headache as a socially invisible disease, psychological processes mediating impaired quality of life, and the process of medication overuse. Initial thematic analysis and subsequent synthesis gave three overarching themes: 'headache as a driver of behaviour' (directly and indirectly), 'the spectre of headache' and 'strained relationships'.

**Conclusion** This meta-synthesis of published qualitative evidence demonstrates that chronic headaches have a profound effect on people's lives, showing similarities with other pain conditions. There were insufficient data to explore the similarities and differences between different chronic headache disorders.

## Strengths and limitations of this study

► As far as we are aware, this is the first systematic review concerning patients' views of chronic headache.
► Although the numbers of studies included in this review are small, it was possible to carry out a meta-ethnographic synthesis.
► Descriptors of study participants' characteristics map poorly into the International Headache Society International Classification of Headache Disorders (ICHD-II) classification of headache chronicity.

in people with frequent acute headaches who take analgesic or specific antimigraine compounds (eg, triptans) on ≥10–15 days per month. We have used a definition of chronicity as a headache occurring on 15 or more days per month for more than 3 months taken from the International Headache Society (IHS) classification guidelines (International Classification of Headache Disorders (ICHD-II)).[3] Chronic headaches have high personal and financial costs[4 5] and have commonly escalated from an episodic presentation.[6 7] This transition from intermittent to chronic headache is associated with an amplified impact on people's lives,[8 9] although the transition between episodic and chronic headache is not always unidirectional or 'fixed'.[2]

The WHO has highlighted the financial burden that headache disorders have on economies at all levels and, paradoxically, the lack of importance generally attributed to these conditions.[10] The usual treatment of chronic headache is pharmacological—either to relieve or prevent attacks.[11 12] The importance of the non-pharmacological management of chronic pain in other sites is recognised,[13] and similar approaches may have a role in chronic headache. In order to develop such non-pharmacological approaches to management, it is important

## BACKGROUND

Viewing chronic headache from a patient's perspective can give insights rarely explored by other research methodologies. This systematic review highlights the need for qualitative input into the little researched area of chronic headache, which has a population prevalence of around 3%–4%.[1 2] We have focused on the the most common causes of chronic headache: chronic migraine (CM), chronic tension-type (CTT) and medication overuse (MO), or a combination of these. Tension-type and migraine are primary headaches. MO headache is a secondary headache that can develop

to understand the lived experience of those living with chronic headaches. This review of the qualitative literature aimed to explore patients' experiences of living with chronic headache and which aspects of living with chronic headache have the most influence on their quality of life.

## METHODS

For this review we were interested in the experience of people with chronic headache. Our population of interest was those with CM, CTT and MO headaches, excluding cluster headaches and other causes of secondary headaches. Recognising the changes in terminology over time, we included any paper where at least half the subjects were reported to have chronic headache affecting them for more than half the days, however defined.

### Searches

The protocol for this systematic review is published in the international prospective register of systematic reviews on 9 September 2015, registration number CRD42015025891 http://www.crd.york.ac.uk/PROSPERO.

We used electronic database searches and hand searches of relevant journals. All electronic searches set limits on human studies and between January 1988 (the date of the publication of the first 'International Classification of Headache Disorders (ICHD)' by the IHS) and July 2016.

One reviewer (VPN), with the assistance of an academic librarian, searched six electronic databases: Medline, Embase, Applied Social Sciences Index and Abstracts (ASSIA), PsycINFO and Scopus. We also conducted forward citation searches on seminal studies in the Web of Science (Science Citation Index and Social Science Citation Index).

The electronic search terms included Medical Subject Headings (MeSH) headings, text words and truncation. Full details of the Medline search (as an exemplar) are available in online supplementary appendix 1.

The same reviewer (VPN) hand-searched nine relevant journals for any relevant studies: *Cephalalgia, British Journal of General Practice, Family Practice, Headache: The Journal of Head and Face Pain, International Journal of Qualitative Studies on Health and Well-being, Patient Education and Counselling, Qualitative Health Research,* and *Qualitative Research in Sport, Exercise and Health.*

### Inclusion criteria

► Studies of chronic headache, including migraine headache, tension-type headache, MO headache or a mixture of these conditions (where at least half the subjects were reported to have chronic headache affecting them for more than half the days, however defined). We accepted the original authors' definitions of chronicity and headache type.
► Qualitative studies from a patients' perspective involving adults (≥18 years) reporting patient experience.

► Qualitative studies with mixed perspectives (eg, healthcare providers or carers and patients), or of mixed quantitative and qualitative methods, only if they analysed and presented the qualitative data on patient experience separately.

### Exclusion criteria

► Studies of people with cluster headaches and all secondary headaches, for example, following a head injury or brain tumour.
► Methodological or theoretical papers, unpublished theses and dissertations, and conference abstracts.
► Non-English-language studies (apart from French, German and Spanish publications).

### Identifying chronic headache studies in the retrieved literature

We had originally intended only to include studies that defined chronic headache according to the IHS ICHD-II guidelines (15 or more headache days a month for 3 months).[3 14] In practice, it appears that authors of qualitative papers rarely use this definition, using a variety of other terms. The term 'chronic headache' has also been used by some authors to mean an episodic headache recurring over a prolonged period; some studies use the term 'frequent headache'; some studies do not make a distinction between episodic and chronic headache; while others describe 'attacks per month' (especially in migraine), rather than supplying actual or average headache days per month. In this review we examined each study to ascertain if they acknowledged chronic headache in any way. When studies described the exact frequency of headaches, a judgement was made as to whether they contained any of our population of interest. When no frequency or chronicity was reported, studies were scrutinised by two to three reviewers to see if half their participants could have had chronic headache and a judgement was made among the team about study inclusion. Our aim was to identify studies where at least half study population had headaches that were compatible with the IHS ICHD-II definition.[3 14]

### Screening

Two reviewers (VPN and AK) independently screened the abstracts and titles against the eligibility criteria. After removing duplications, full texts of potential studies were then screened by the same reviewers. A further two reviewers (ST or DE) verified any disputed or ambiguous studies and agreed the final eligible studies.

Two reviewers (VPN and SS see acknowledgements) assessed the quality of the papers of interest using an adaptation of the Critical Appraisal Skills Programme (CASP) quality assessment tool.[15] A third reviewer (ST) adjudicated disagreements. Our intention was not to exclude studies based on quality but to consider the quality assessments while interpreting the study findings and generating our conclusions.

### Analysis

We used a thematic synthesis approach[16] for the initial analysis, and for subsequent synthesis of the data we

used a meta-ethnographic approach.[17 18] Currently there is no detailed published guidance on how to conduct a meta-ethnographic synthesis, although work is being carried out to standardise this process.[19] Meta-ethnographic synthesis commonly describes three orders (or levels) of constructs (also known as themes or concepts): first-order constructs reflect the understanding or lay interpretation of the research participants and are often found in direct quotes in the result sections of included papers; second-order constructs are the interpretation of participants' understandings made by the authors of the primary studies; third-order constructs arise from synthesis of the first-order and second-order constructs—usually across different studies—to form new theories or models.[20] In practice it is often difficult for reviewers to distinguish between first-order and second-order constructs included in primary studies.[20]

Four members of the team (AK, VPN, ST and DRE) read the studies and discussed the similarities and differences between them. The results and discussion sections of the included papers were then coded line by line using NVivo V.10 software (QSR International, 2012), as described by Thomas and Harden,[16] and grouped into themes. New themes were developed as required. This new set of themes and the relationships between the themes were explored and refined into our third-order conceptual themes.

Noblit and Hare[18] describe three types of meta-ethnographic synthesis of studies: reciprocal translation when they are about similar things, refutational when they refute each other and lines of argument when successively they build a line of argument. We attempted to synthesise the studies meta-ethnographically using the process of reciprocal translation (the most appropriate of the three approaches for our data), which involves taking the first study data and mapping a second study onto it, noting any exceptions and omissions (ie, study 2 is similar to study 1 noting exceptions). This continues with each study.

We explored the studies chronologically mapping the data to the third-order themes. The studies were reread and initial study concepts extracted under the new third-order conceptual headings to make comparisons of these themes across the studies.

## RESULTS
### Searches
Electronic and hand searches identified 6421 potentially relevant citations, which included 3586 unique citations (see flow chart in figure 1).

Overall we identified 86 studies that warranted scrutiny of the full text. Exclusions are detailed in figure 1. Five studies that contained some people with chronic headache were excluded because they did not meet all the study criteria.[21–25] Further details are given in online supplementary appendix 2. They are not formally included in the synthesis. Although our inclusion criteria stated English, Spanish, German and French studies only, our searches did not identify any other languages.

### Studies included and quality appraisal
Four studies met the inclusion criteria and were included in the review.[26–29] We quality-appraised the eligible studies using an adapted CASP tool.[30] Quality is depicted in table 1 using 'Yes' if they met the criteria, 'unclear' where it was unclear whether they met the criteria or 'No' if they did not meet the criteria.

For characteristics of the included studies, see table 2 and table 3.

All the studies were based in different countries: USA, Sweden, Italy and Finland/UK. There were 73 participants, 52 of whom were female; their ages ranged from 22 to 82. Study recruitment varied from a tertiary clinic,[26] advertisements,[27] a headache centre[29] and a snowballing technique where it was unclear how the first patients had been identified.[28]

Coeytaux et al[26] used focus groups, and the other three studies used face-to-face, individual interviews. Coeytaux et al do not explicitly state their approach to qualitative analysis but describe processes consistent with thematic analysis. The other three interview studies used a grounded theory approach.

### Headache characteristics
Headache types, when specified, were either; medically diagnosed (Coeytaux et al, Lonardi, and Tenhunen and Elander) or self-reported (Jonsson et al). Across all four studies they included people with migraine (31), tension-type headache (13), cluster headache (2), daily persistent headache (18), and chronic daily or near daily headache (9). Some participants had more than one type of headache.

The chronic headache definitions included differed slightly across the studies but all were compatible with the IHS classification (see table 2).

The four papers had very different aims: Coeytaux et al explored participants' views of outcome measures and the assessment of meaningful change, with a view to informing clinicians about patients' perspectives; Jonsson et al explored the lives of people with MO headache in order to better understand the development of this condition; Lonardi focused on acceptance by social groups, headache as an invisible disease and the passing dilemma (ie, do patients disclose or hide their condition?); and Tenhunen and Elander identified psychological processes that mediate impaired quality of life.

### Thematic synthesis
In table 4 we present the themes and findings from the four included studies that contain first-order and second-order themes. First-order themes are data from the participants of the studies, and second-order themes are those that the authors have concluded, although there are potential overlaps with these data.[19]

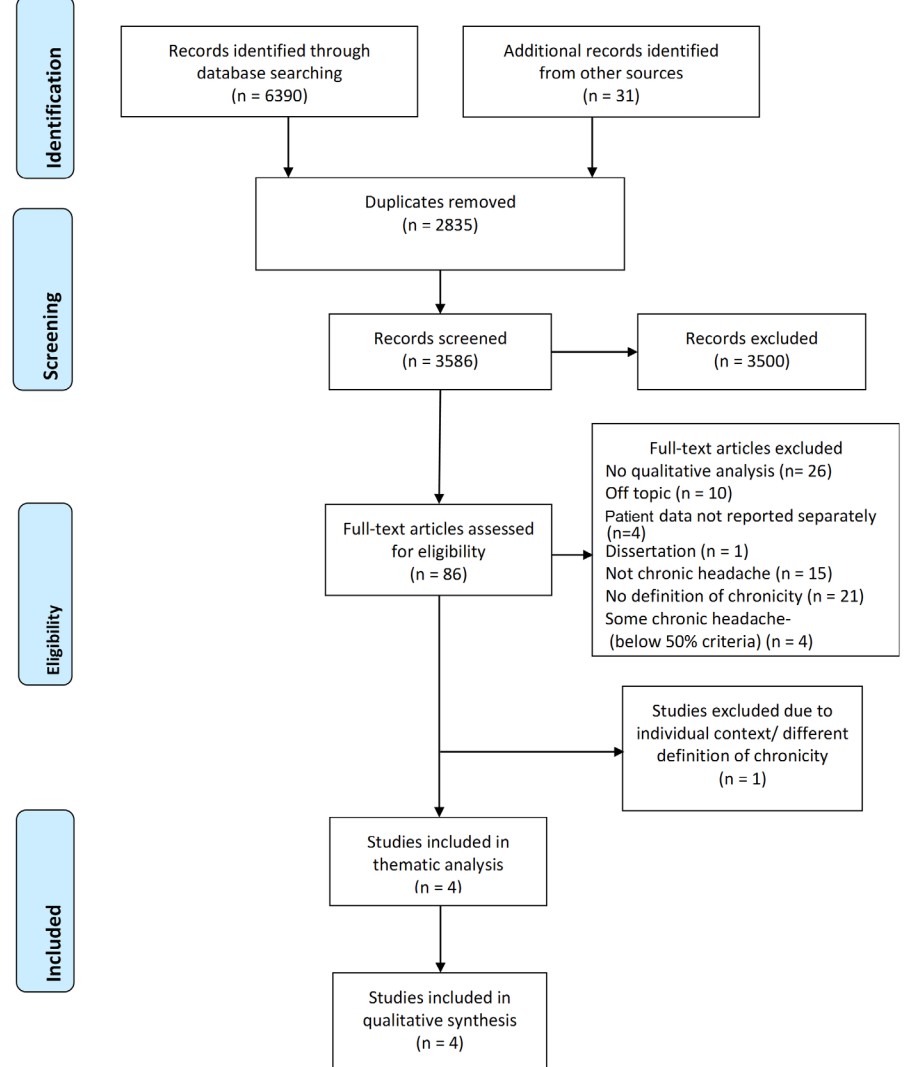

**Figure 1** Preferred Reporting Items for Systematic Reviews and Meta-Analyses (PRISMA) flow diagram.

## Meta-ethnographic synthesis

Here we describe the results of our synthesis. We coded the studies' results and discussions, which gave rise to new themes that include first-order and second-order themes.

Figure 2 shows how these themes were explored and refined into three new emergent themes.

Online supplementary appendix 3 shows how these themes were synthesised across the papers chronologically using reciprocal translation.

We use some exemplary quotes in the text but have included the full data used in online supplementary appendix 4. First-order and second-order themes are difficult to separate, but for the purposes of clarity participant quotes from the studies are italicised and any authors' interpretations are given in normal text. Online supplementary appendix 5 shows theme saturation across the studies and how some papers contributed more to the data than others.

### Headache as a driver of behaviour
#### Direct

Participants described headache as a 'driver' leading to behaviours such as increased medication use, avoidance of planning, change in sleep patterns or having to stop doing things. All of these lead to a feeling of a loss of control in their lives. The headache is described as directly leading to these behaviours and we considered these to be direct effects.

*"…I lost my freedom to plan what to do tomorrow…"* (Lonardi, p1622).[29]

*"I don't have as much control over my life as my husband has…""I have to do whatever my illness requires and allows me to do"* (Tenhunen, p403–404).[28]

'Taking medication because one has to, not because one chooses to' (Jonsson, p7).[27]

#### Indirect

There were also indirect or 'knock on' effects, such as financial difficulties due to their performance in the workplace being affected, or changes in the type of job they could do in order to accommodate their headaches. Sleep was also affected due to their headache or medication use.

**Table 1** Quality appraisal

| Question | Coeytaux et al,[26] | Jonsson et al,[27] | Lonardi,[29] | Tenhunen and Elander,[28] |
|---|---|---|---|---|
| 1. Is this study qualitative research? | Yes | Yes | Yes | Yes |
| 2. Are the research questions clearly stated? | Yes | Yes | Yes | Yes |
| 3. Have ethical issues been taken into consideration? | Yes | Yes | Unclear | Unclear |
| 4. Is the qualitative approach clearly justified? | Yes | Yes | Yes | Yes |
| 5. Is the approach appropriate for the research question? | Yes | Yes | Yes | Yes |
| 6. Is the study context clearly described? | Yes | Yes | Yes | Unclear |
| 7. Is the role of the researcher clearly described? | Yes | Yes | No | No |
| 8. Is the sampling method clearly described? | Yes | Yes | Unclear | No |
| 9. Is the sampling strategy appropriate for the research question? | Unclear | Unclear | Yes | Unclear |
| 10. Is the method of data collection clearly described? | Yes | Yes | Unclear | Unclear |
| 11. Is the data collection method appropriate to the research question? | Yes | Yes | Unclear | Unclear |
| 12. Is the method of analysis clearly described? | Yes | Yes | No | Unclear |
| 13. Is the chosen analytical approach suitable for addressing the research question? | Yes | Yes | Yes | Yes |
| 14. Are the claims made supported by sufficient evidence? | Unclear | Yes | Yes | Yes |

'They also thought that their headaches would affect more long-term factors such as wages and pensions and that they could eventually force them to choose a less demanding job or even early retirement' (Jonsson, p5).[27]

"…I have to wake up several times a night to take some more medication and the regular awakenings are pretty disturbing" (Tenhunen, p401).[28]

'Social exclusion leads to losing personal value as a human resource, experiencing related feelings of exclusion, isolation and loneliness' (Lonardi, p1623).[29]

### The spectre of headache

The spectre of headache looms large in the data from these studies. The studies describe participants' concerns about planning, fears about their pain, worries about medication and perceived lack of control about whether strategies would work in the future. Participants also felt guilty about burdening others. We have used the word spectre as a nebulous but potentially menacing ever-present 'cloud of concern' that patients have to take into account with all relationship transactions and forward planning.

'Sometimes they even avoided making appointments because they dreaded having to cancel them' (Jonsson p5).[27]

"…I'm scared that something uglier is there but there's nothing…my worry is always the same but when I see that all the tests are negative, then I wonder why I have a headache" (Lonardi, p1623).[29]

"These triptans are the only thing that I have found that really helps, so that I can live my life and do what I want to…if it stops or if I'm not allowed to take it any more…Just thinking about it makes me very nervous" (Jonsson, p4).[27]

### Strained relationships

Strained relationships, the final theme, concerns the impact patients' headaches have on other people and their relationships with them and their communities.

More often than not this will be family and close friends, although work colleagues can also be affected. Relationships are also affected by the person being unable to plan. Although some people spoke of some close relationships being helpful when headaches seem to take control of peoples' lives, this can cause anxiety and can have an impact on how they view themselves. The theme of strained relationships shows that other people are deeply affected by this condition and that the relationship issues do not necessarily remain constant. Other peoples' attitudes may change from being supportive, overprotective or undermining. The person with chronic headache has to make personal decisions about disclosure, which may alter in different contexts or scenarios. They try to act normally to lessen the chance that people will be critical or dismissive of their condition.

'…each group reported strained relationships resulting from the need to change or cancel plans because of headaches' (Coeytaux, p483).[26]

"I used to be the strong one in the family…but now when I'm not so strong people tend to save me from hearing bad news…" (Tenhunen, p402).[28]

'Sometimes they sensed that other people were suspicious, presumably thinking that headaches were being used as an excuse' (Jonsson, p6).[27]

**Table 2** Core studies characteristics 1

| Citation, country | Design | n=M/F, age range | Headache type | Definition of chronicity | Sources of recruitment |
|---|---|---|---|---|---|
| Coeytaux et al,[26] USA | Focus groups | 19 (5M/14F), 22–83 years | 68% (13) diagnosed with migraine by headache clinic neurologist, 32% (6) with tension-type headache, and 32% (6) had one or more IHS headache diagnoses | 15 or more days with headache in month preceding enrolling in RCT, duration of chronic headache pain 4 to >60 years, all but one had experienced pain-free periods ranging from a few days to a month or more | Recruited from a university-based tertiary care headache clinic who had recently participated in an RCT |
| Jonsson et al,[27] Sweden | Individual qualitative interviews | 14 (5M/9F), 36–64 years | Diagnosis of medication overuse headache (2006 IHS criteria); 10 participants self-reported migraine, mostly in combination with tension-type headache; 4 reported only tension-type headache | All reported having daily or near daily headaches. | Recruited through advertisements in the national journal of a headache patient organisation and local daily newspaper |
| Lonardi,[29] Italy | Open-ended biographic interviews | 31 (7M/24F), 23–74 years | All primary chronic headache; cluster=2, migraine=8, TTH=3, other primary headache (daily persistent headache)=18 | Introduction talks about chronic headache, no definition given; characteristics table by type of headache; daily persistent headache not defined; two quotes talk about daily headache | Recruited from headache centre, randomly selected by a convenience sample of consecutive patients |
| Tenhunen and Elander,[28] Finland and UK | Semistructured interviews | 9 (4M/5F), 32–55 years | Clinical diagnosis of chronic daily or near daily headache | Headaches occurring on 15 or more days per month lasting for more than 4 hours per day | Purposive theoretical sampling to compare with those in previous quantitative studies (unspecified) of CDH; used a snowballing technique; no mention of where or how or why across two countries |

CDH, chronic daily headache; IHS, International Headache Society; RCT, randomised controlled trial; TTH, tension type headache.

**Table 3** Core characteristics 2

| Citation | Aims of study | Qualitative methods/ analysis | Interview questions |
|---|---|---|---|
| Coeytaux et al,[26] | To identify which clinical outcomes are most important to patients with frequent headaches; to inform clinicians which of the many available headache assessment instruments may be helpful to assess meaningful change over time from the patient's perspective | Focus groups | Participants were asked to (1) describe what their headaches were like and how their headaches affected their day-to-day activities; (2) describe other symptoms or sensations usually accompanying their headaches; (3) indicate some of the signs or indicators that their headaches were lessening; (4) delineate which of those symptoms from which they most wanted to get relief. They were asked if they thought quality of life was something separate from their pain severity. They were given copies of questionnaires completed during the study and asked which ones were helpful to determine whether they got better over time. |
| Jonsson et al,[27] | To explore how individuals with medication overuse headache use medications and other strategies to manage headaches in their daily lives, and their thoughts about their own use of acute medication | Grounded theory, interviews, | The opening question was 'Could you tell me about your headaches?' Participants were asked questions about their headaches and daily life, strategies to manage headaches, use of medication and thoughts about using less medication. They were encouraged to tell their stories freely and probing questions were used to obtain as much detail as possible. |
| Lonardi,[29] | To explore the chronic headache experience as an invisible disease through narrative reconstruction | Grounded theory, interviews | A loose interview guide from clues collected from the first patients (in abstract)—asked how participants became aware of the disease, illness and sickness aspects of their headaches |
| Tenhunen and Elander,[28] | To gain insight into the psychological processes which mediate quality of life impairments in chronic daily headache (CDH) | Grounded theory, interviews | Asked about the impact of CDH on occupational, physical, psychological and social areas of life |

**Table 4** Study themes

| Coeytaux et al,[26] | Jonsson et al,[27] | Lonardi,[29] | Tenhunen and Elander,[28] |
|---|---|---|---|
| **Five salient topics:** | **Process of medication overuse (with subthemes):** | **Illness, disease and sickness; headache representations** | **Seven categories of quality of life impairment:** |
| ► Pain severity | *Headaches threaten to ruin one's life* | Four scenarios: | ► Daily activities |
| ► Meaningful symptom relief | ► Headaches are unbearable | ► Fully accepted by family | ► Work and education |
| ► Uncertainty | ► An extra burden in everyday-life | ► Partial acceptance by family | ► Sleep energy and concentration |
| ► Devaluation | ► Having to make life adjustments | ► Unaccepted by family, work expulsion from the productive world | ► Social activities |
| ► Meaningful assessment | ► Struggling to be able to work | ► Accepted by inner family but stigmatised/excluded from the outer world | ► Emotional reactions |
| *Other findings:* | ► Being forced to cancel important events | People trivialise the condition | ► Perceptions of self |
| ► The most important outcome measures to patients are pain severity and frequency. A meaningful measure would be a change in pain-free days. | *Medication as the only solution* | **Invisible disease and the passing dilemma** | ► Effects on partners and family |
| ► Disability, quality of life and functional measures were not as important to patients. | ► Searching for explanations | Whether to suffer in silence or talk about the condition | *Core category*—reduced control either perceived or actual loss of control; increases vulnerability |
| ► Diaries were not burdensome and helped to identify trends over time. | ► Testing numerous strategies | | |
| | ► Scepticism towards prophylactic medication | | |
| | ► Resignation: nothing but the medication helps | | |
| | ► Always having the medication at hand | | |
| | *Short-sighted medication use* | | |
| | ► Taking the medication because one has to, not out of choice | | |
| | ► Focusing on the headache when deciding whether to medicate | | |
| | ► Avoidance of tracking medication use | | |
| | ► Increased medication use during stressful periods in life | | |
| | ► Perceptions about the link between increased headaches and medication use | | |

There is a great deal of overlap between these themes. When the headache acts as a driver of behaviour, it affects the relationships with those around them.

'…headaches placed stress on relationships not only by affecting their own behaviour but also because of the confusion, frustration and fear experienced by their partners:' (Tenhunen p402)[28]

This effect on relationships can also fuel the worries and fears noted in the theme of the spectre of headaches.

*"Sometimes I also feel like a trouble to the others. Dependent on everybody." "I think that I'm causing a lot of extra work for my wife…"* (Tenhunen, p402).[28]

CTT headache, although represented (n=13) in all studies except Tenhunen and Elander, is only mentioned once by Jonsson *et al*, who commented that people with CTT headache were less distressed than participants with migraine.

'Some were afraid of the pain, afraid of the next attack. Those who had tension type headache as the primary headache described the pain as disturbing rather than frightening.' (Jonsson p4,5)[27]

## DISCUSSION

Our meta-ethnographic synthesis identified three overlapping, overarching effects of chronic headache on patients' lives: as a driver of behaviour, as a spectre hanging over lives and a cause of strain on relationships.

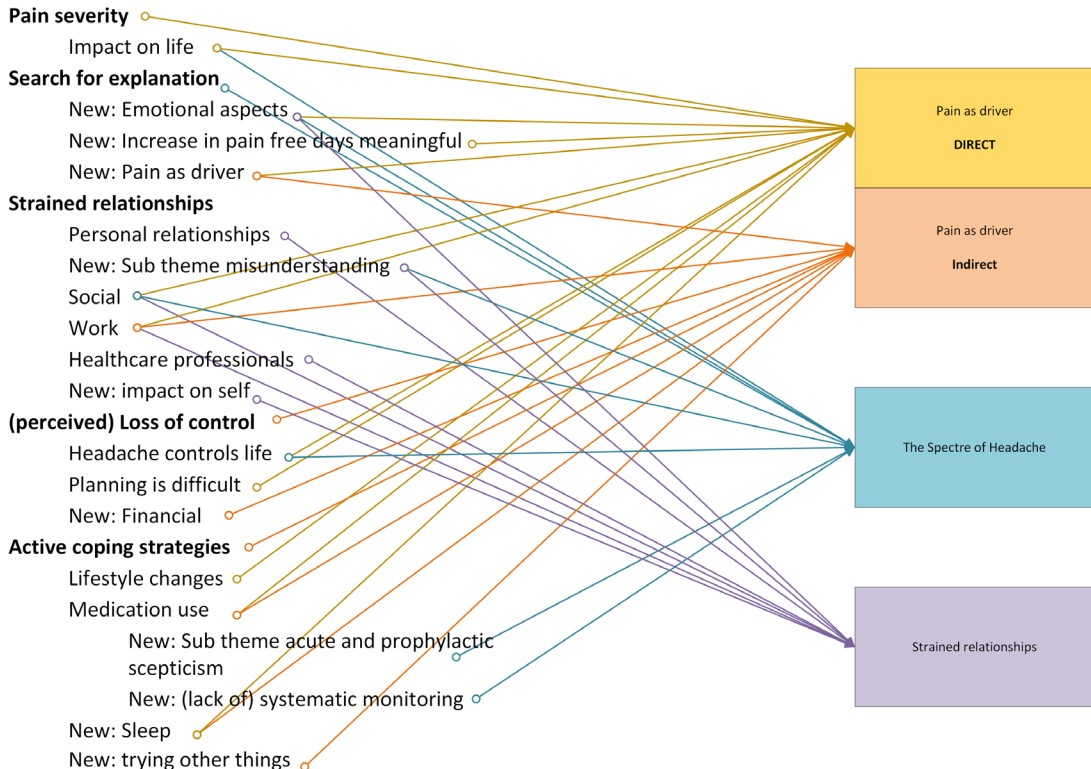

**Figure 2** Third-level conceptual framework.

Our findings show that headache can cause direct and indirect effects on peoples' lives, the uncertainty of when a headache will strike leaving them unable to plan and forcing them to prioritise their lives around their headaches. The spectre of headache is an ever-constant cloud of concern containing their worries and fears, bringing feelings of a loss of control and guilt. As headaches increase in frequency, it becomes more difficult for people to function as headaches become a driving force affecting their lives and relationships.

We have followed a rigorous search strategy and are confident that we have kept the search wide enough to capture the salient literature. Nevertheless, the number of studies is very small, and this may mean that important themes may not have been identified simply because the spectrum of research studies is too narrow. For example, although tension-type headache was represented in the studies, very little was written about whether the impact on lived experience differed from migraine, apart from the one quotation suggesting that these patients found their headaches 'disturbing rather than frightening'. It is unclear whether this meant that the pain was perhaps less severe or was less distressing. We have followed an analysis plan with an experienced qualitative team, which can be replicated by others. Although we attempted as rigorously as possible to identify studies where at least 50% of those studied has chronic headache compatible with the ICHD-II classification,[3 14] because of the limited and varied descriptions of participant characteristics, we

may have inadvertently excluded some studies where this was not the case.

Our themes are very similar to the views of participants in the qualitative studies of episodic migraine literature, which describe feelings of a loss of control, impaired quality of life, the impact on family, work and social relationships, stress of the unpredictability, and worry that there may be a tumour causing their headaches.[31–36] Perhaps this is unsurprising as most CM headaches start with an episodic presentation.[6]

Many of the issues raised in this review such as impaired quality of life, the effect on others, uncertainty and the emotional impact of the condition resonate with other chronic pain conditions such as low back pain (LBP).[37 38] LBP and headache could both be described as being common disorders of the pain matrix, which are often devalued by others.[39] Lonardi also points out that people who have not experienced chronic headache are likely to have experienced a headache and may feel they understand what a 'bad headache' is; they may even use the term 'migraine' for a bad headache. This is very different from other long-term conditions, for example, epilepsy, as you cannot experience 'a bit of' this condition.[29]

Chronic pain conditions are known to respond to psychological approach treatments, whereas treatments for chronic headache are mainly by medication only. Those with CM show a similar pattern of disability and distress to episodic migraine, although at a greater frequency, which may suggest an increased burden.

Our four studies highlight chronic headache as a distressing, invisible, under-recognised condition that needs further research. The literature at present may not fully represent all aspects of chronic headache.

## CONCLUSIONS AND RECOMMENDATIONS FOR FUTURE RESEARCH

Chronic headaches have a profound effect on people's lives, showing similarities with other pain conditions, and may benefit from psychological approaches used in chronic pain management. Future research is needed to explore patients' perspectives of the evolution of chronic headache and how they view strategies to address MO. It is unclear whether tension-type headaches were truly represented here or whether the features of migraine may have overshadowed the data. This is an area worthy of further exploration.

**Acknowledgements** Due to the use of extensive quotes in our analysis we have obtained permissions from the four publishers of our included papers; John Wiley and Sons reference 26, SpringerOpen reference 27, Elsevier reference 29 and SAGE publications reference 28. We would like to thank Saseela Subaskaran, who helped with the CASP appraisal, and Samantha Johnson, the academic librarian who helped with the electronic searches.

**Collaborators** Felix Achana, Division of Health Sciences, Warwick University; Mary Bright, University Hospitals Coventry and Warwickshire; Fiona Caldwell, Royal Holloway University of London; Dawn Carnes, Queen Mary University London; Brendan Davies, Royal Stoke University Hospital; Sandra Eldridge, Queen Mary University London; Simon Evans, Migraine Action; Kirstie Haywood, Division of Health Sciences, Warwick University; Siew Wan Hee, Division of Health Sciences, Warwick University; Helen Higgins, Warwick Clinical Trials Unit; Manjit Matharu, National Hospital for Neurology & Neurosurgery; Hema Mistry, Division of Health Sciences, Warwick University; Shilpa Patel, Warwick Clinical Trials Unit; Stavros Petrou, Division of Health Sciences, Warwick University; Tamar Pincus, Royal Holloway University of London; Rachel Potter, Warwick Clinical Trials Unit; Katrin Probyn, Royal Holloway University of London; Harbinder Sandhu, Warwick Clinical Trials Unit; Wendy Thomas, The Migraine Trust White; Kimberly Warwick, Clinical Trials Unit.

**Contributors** ST, MU and DRE contributed to review concept and design. VPN and AK screened all search results, independently coding records for inclusion/exclusion, then extracted data from included studies. VPN and ST conducted the analysis and synthesis. VPN, ST, FG, DE and MU contributed to data interpretation. VPN prepared the manuscript, and all authors revised it critically for important intellectual content and approved the final manuscript.

**Funding** This work was supported by the National Institute for Health Research Programme Grants for Applied Research (Chronic Headache Education and Self-management Study (CHESS) ISRCTN Number: 79708100). The views expressed are those of the author(s) and not necessarily those of the NHS, the NIHR or the Department of Health.

**Competing interests** MU and ST report grant from National Institute for Health Research. MU reports personal fees from National Institute for Health and Care Excellence, grants from Arthritis Research UK, personal fees from National Institute for Health Research, outside the submitted work, and was Chair of the guideline development group that produced the 2012 NICE headache guidelines. The authors declared no potential conflicts of interest with respect to the research, authorship and/or publication of this article.

**Ethics approval** Ethics approval was provided by West Midlands – Black Country Research Ethics Committee (15/WM/0165).

**Provenance and peer review** Not commissioned; externally peer reviewed.

**Data sharing statement** No additional data are available.

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
