## [Reviewer comments · BMJ Open]

ARTICLE DETAILS

TITLE (PROVISIONAL)	The lived experience of chronic headache: A systematic review and synthesis of the qualitative literature
AUTHORS	Nichols, Vivien; Ellard, David; Griffiths, Frances; Kamal, Atiya; Underwood, Martin; Taylor, Stephanie

VERSION 1 - REVIEW

REVIEWER	George Bramley Institute of Applied Health Research University of Birmingham United Kingdom
REVIEW RETURNED	06-Apr-2017

GENERAL COMMENTS	This clearly well thought through and executed qualitative synthesis of individuals' experiences of chronic headaches. The methodology is clearly stated as are pragmatic decisions that were taken during the execution of this review. The authors clearly set out the challenges encountered including the different foci, methodologies and analysis approaches adopted by the four included papers. While the primary studies included would have ideally selected and described participants according to IHS classification this should not detract from the important insights this study provides. Particular strengths of this paper include: (a) well written introduction that sets the context and need for the study; (b) being underpinned by a comprehensive literature search and input by experienced qualitative researchers; (c) the concise and comprehensive description of the review methodology; (d) reporting of the analysis with supporting tables in the appendices that allow the reader to map back to the original material. This paper provides a useful baseline on currently available evidence on patients' experiences of living with chronic headaches and the recommendations for further research are sound. Specific points P.; line 6: I was not able to identify any other similar review p.3; line 15: Description of patients, or lack of, may simply be a reflection of the original purpose and intended audience of the included papers. p.5, line 8: acronym MOH not previously introduced p.5, line 28: January 1988 is a reasonable start date and clear rationale given. p.6, line 9: missing word 'included' after 'were'? p.6, line 35 onwards: clearly sets out the issues in selecting studies and the pragmatic approach adopted to selecting studies for inclusion. p.7, line 29 onwards: good accessible description of synthesis approach adopted that clearly identifies methodological issues for the reader.
--

	p.10, first paragraph. Did the team consider whether there were issues related to combining data from focus groups with data from individual interviews? p.17, Discussion identifies and addresses limitations of the study. Figure 2 – development of third order codes – will this be in colour to aid the reader? Some page numbers for supporting quotes in appendix are missing.
--	--

REVIEWER	Roy Beran University of NSW, Australia
REVIEW RETURNED	25-Apr-2017

GENERAL COMMENTS	Referee's Report "The Loved Experience of Chronic Headache: A systematic review and synthesis of qualitative literature" Manuscript ID: BMJ Open-2017-16787 This is a qualitative study, which to many clinicians, is a less well-accepted approach because it lacks the hard quantity data that is so basic to scientific research other than qualitative work. One of the initial limitations is the exclusion of non-English language studies when there are so few studies available. The authors, in the Methodology, initially wanted to use the International Headache Society 1988 definitions, which have long been superseded and they acknowledged that even this proved impossible. A lack of consistency and even a proper definition make this paper difficult to accept. It is impossible to know whether each of the papers reviewed looked at exactly the same basic diagnosis and the material, as provided by the authors, would suggest that this is not the case thereby making any interpretation dubious. The statement, "... When no frequency or chronicity was reported, studies were scrutinised by 2 - 3 reviewers to see if half their participants could have had chronic headache...", is largely a nonsense and makes the whole methodology so inconsistent as to make the paper less than acceptable. The terms "chronic headache" and "chronic daily headache" are not synonymous and to accept them as such is to defeat the purpose behind the studies. The Methodology is a composite of a number of methods resulting in a new approach, which of itself is interesting, especially to scientists not well versed in qualitative research. In the end, there were 4 studies from vastly different environments with different cultures and prevailing circumstances resulting in very small numbers, comprising 73 disparate participants with equal gender participation and a vast age range from 20 to 80 years with different recruitment methods. The whole issue of meta analysis with such a population is highly dubious and to draw conclusions from such meta analysis seems unacceptable. Even the classification of headaches was disparate with some being self diagnosed by the participants and others clinically diagnosed, making the whole meta analysis highly dubious. The authors point out that "... The 4 papers had very different aims..." and yet they
--

	chose to combine them for a meta analysis and I do not believe that that is appropriate. It seems a little like combining apples and oranges because they are both fruit and then giving a dissertation on an analysis based on fruit, disallowing any other differences. - 2 - The sentence, "... The spectre of headache looms large in the data from these studies...", appears a nonsensical statement if the 4 studies were based on headache and if this is the sort of interpretation that the results provide, then it suggests an over interpretation or analysis approach, reiterating my previous analogy of fruit for oranges and apples. The opening paragraph of the Discussion section failed to acknowledge the possibility of the "chicken and the egg" effect, putting all the emphasis on headaches causing the problems rather than the problems causing headaches. This issue has been totally overlooked by the authors and the fact that headaches may be an excuse for other behaviour has also been totally ignored by the authors. The interpretation as provided, smacks of a bias underpinning much of this research, which ignores the fact that it may be life factors that are causing headaches rather than the other way around. The authors acknowledge the "very small" number of studies reviewed and the possible bias within research and they further acknowledge that the language used was "unclear", with the meaning being ambiguous. The authors did state that they attempted to be rigorous in their methodology but even they questioned their own interpretation. I could go on in my criticisms of the paper but will stop at this point to say that I do not believe the paper is suitable for publication within the journal and I cannot see how the authors can revisit the work to make it acceptable. Such a paper may have greater value to a Social Sciences journal rather than a journal directed towards clinicians, particularly involved with the management of headache.
--	---

VERSION 1 – AUTHOR RESPONSE

Reviewer 1

We thank you very much for your intuitive and kind remarks about our paper. In the table below we have responded to your specific queries and comments.

Reviewer 1 comments Response

P.; line 6: I was not able to identify any other similar review Thank you for your confirmation of this
p.3; line 15: Description of patients, or lack of, may simply be a reflection of the original purpose and intended audience of the included papers.

Thank you for this observation we agree.

p.5, line 8: acronym MOH not previously introduced We have amended this by adding text

p.5, line 28: January 1988 is a reasonable start date and clear rationale given.

We agree

p.6, line 9: missing word 'included' after 'were'? We have reworded the text for clarity

p.6, line 35 onwards: clearly sets out the issues in selecting studies and the pragmatic approach adopted to selecting studies for inclusion.

We agree – thank you

p.7, line 29 onwards: good accessible description of synthesis approach adopted that clearly identifies methodological issues for the reader.

We agree – thank you

p.10, first paragraph. Did the team consider whether there were issues related to combining data from focus groups with data from individual interviews?

You raise an interesting point. As we were primarily dealing with patient data and generated themes we did not feel that we needed to make this demarcation.

p.17, Discussion identifies and addresses limitations of the study.

We agree – thank you

Figure 2 – development of third order codes – will this be in colour to aid the reader?

An excellent idea, thank you. We have updated this; we hope this is acceptable and helps.

Some page numbers for supporting quotes in appendix are missing. We have checked and amended this

Reviewer 2

Response

The team would like to thank the reviewer for taking the time to review our paper and providing us with such frank feedback.

We have considered the issues you raised about the use of diagnostic groups but feel we have covered this issue sufficiently in the paper.

We note the concerns about the methodology and its appropriateness in medical literature. Meta-ethnography is a well-established methodology. It has a history going back to the 1980s when the first major text on this methodology was published in the education field (Noblit & Hare 1988). Since this date it has developed considerably and is now commonly used in healthcare reviews and there is a growing number of published qualitative meta-ethnography papers related to health. To illustrate this we provide a number of examples at the end of the document. We feel that papers like these are of interest to a clinical as well as research audiences. Although we only found four papers this does not detract from the method, indeed it is a telling result. The aim of the study was to find out what people with chronic headache experience. From the literature it seems they have rarely been asked. The lack of findings are perhaps as important as the findings in this piece of work. It is of interest that four such different studies produced overlapping themes and could be synthesized.

RE comment on English language papers: Thank you for your comment. We did include studies in French, German and Spanish to be as inclusive as possible with the access we had to researchers who spoke these languages fluently. In the searches no other languages came up therefore no foreign language studies were excluded. We have added text in the results section to explain this.

RE comment about headache type: A reductionist approach of isolating individual headache types and applying today's diagnostic criteria would mean we would have no information rather than some. We suggest that the experience of living with chronic headaches is not substantially changed by the underpinning diagnosis. Whilst accepting the chicken and egg issue we need to report what the studies found and as that was not their take on the data we cannot add this to the analysis.

The terms "chronic headache" and "chronic daily headache" are not synonymous and to accept them as such is to defeat the purpose behind the studies.

We have included "chronic headache" and "chronic daily headache" in this review as we aimed to be as inclusive as possible rather than to suggest these were diagnostically synonymous terms. We have

changed the text accordingly.

Examples below.

Campbell, Rona, et al. "Evaluating meta-ethnography: a synthesis of qualitative research on lay experiences of diabetes and diabetes care." *Social science & medicine* 56.4 (2003): 671-684.

Elmir, Rakime, et al. "Women's perceptions and experiences of a traumatic birth: a meta-ethnography." *Journal of advanced nursing* 66.10 (2010): 2142-2153.

Cosco, Theodore D., et al. "Lay perspectives of successful ageing: a systematic review and meta-ethnography." *BMJ open* 3.6 (2013): e002710.

Sinnott, Carol, et al. "GPs' perspectives on the management of patients with multimorbidity: systematic review and synthesis of qualitative research." *BMJ open* 3.9 (2013): e003610.

VERSION 2 – REVIEW

REVIEWER	George Bramley Institute of Applied Health Research University of Birmingham United Kingdom
REVIEW RETURNED	13-Jun-2017

GENERAL COMMENTS	This clearly well thought through and executed qualitative synthesis of individuals' experiences of chronic headaches. The review methodology is clearly described and sound for a qualitative synthesis. The authors have been transparent and clearly set out the pragmatic decisions that were taken during the execution of this review. They have clearly set out the challenges encountered undertaking a study of this nature including the different foci, methodologies and analysis approaches adopted by the four included papers. Using the date that IHS classification was published was sensible approach to limiting the volume of data, though it could be argued that would be inevitable lag in their use in study design and more so for qualitative studies concerned with lived experience of people living with chronic headaches. Ideally the included primary studies would have mapped neatly onto IHS classification as this would have been helpful in drawing out implications for more holistic care. That said this study provides insights into lived experiences of patients with CHs. The authors provide a comprehensive and transparent account of their meta-ethnographic synthesis that will allow others to replicate their approach and helpful provide underpinning themes in the supplementary material. The three themes could be potentially used by clinical teams developing more person centred approach to care for individuals suffering from CH. Particular strengths of this paper include: (a) well written introduction that sets the context and need for the study; (b) being underpinned by a comprehensive literature search and input by experienced qualitative researchers; (c) the concise and comprehensive description of the review methodology; (d) reporting of the analysis with supporting tables in the appendices that allow the reader to map back to the original material. The recommendations for further research are sound.
--

REVIEWER	Roy Beran
-----------------	-----------

	UNSW/GRIFFITH U/Strategic Health Evaluators; Australia
REVIEW RETURNED	04-Aug-2017

GENERAL COMMENTS	This is a qualitative study, which to many clinicians, is a less well-accepted approach because it lacks the hard quantity data that is so basic to scientific research other than qualitative work. One of the initial limitations is the exclusion of non-English language studies when there are so few studies available. The authors, in the Methodology, initially wanted to use the International Headache Society 1988 definitions, which have long been superseded and they acknowledged that even this proved impossible. A lack of consistency and even a proper definition make this paper difficult to accept. It is impossible to know whether each of the papers reviewed looked at exactly the same basic diagnosis and the material, as provided by the authors, would suggest that this is not the case thereby making any interpretation dubious. The statement, "... When no frequency or chronicity was reported, studies were scrutinised by 2 - 3 reviewers to see if half their participants could have had chronic headache...", is largely a nonsense and makes the whole methodology so inconsistent as to make the paper less than acceptable. The terms "chronic headache" and "chronic daily headache" are not synonymous and to accept them as such is to defeat the purpose behind the studies. The Methodology is a composite of a number of methods resulting in a new approach, which of itself is interesting, especially to scientists not well versed in qualitative research. In the end, there were 4 studies from vastly different environments with different cultures and prevailing circumstances resulting in very small numbers, comprising 73 disparate participants with equal gender participation and a vast age range from 20 to 80 years with different recruitment methods. The whole issue of meta analysis with such a population is highly dubious and to draw conclusions from such meta analysis seems unacceptable. Even the classification of headaches was disparate with some being self diagnosed by the participants and others clinically diagnosed, making the whole meta analysis highly dubious. The authors point out that "... The 4 papers had very different aims..." and yet they chose to combine them for a meta analysis and I do not believe that that is appropriate. It seems a little like combining apples and oranges because they are both fruit and then giving a dissertation on an analysis based on fruit, disallowing any other differences. - 2 - The sentence, "... The spectre of headache looms large in the data from these studies...", appears a nonsensical statement if the 4 studies were based on headache and if this is the sort of interpretation that the results provide, then it suggests an over interpretation or analysis approach, reiterating my previous analogy
--

	of fruit for oranges and apples. The opening paragraph of the Discussion section failed to acknowledge the possibility of the "chicken and the egg" effect, putting all the emphasis on headaches causing the problems rather than the problems causing headaches. This issue has been totally overlooked by the authors and the fact that headaches may be an excuse for other behaviour has also been totally ignored by the authors. The interpretation as provided, smacks of a bias underpinning much of this research, which ignores the fact that it may be life factors that are causing headaches rather than the other way around. The authors acknowledge the "very small" number of studies reviewed and the possible bias within research and they further acknowledge that the language used was "unclear", with the meaning being ambiguous. The authors did state that they attempted to be rigorous in their methodology but even they questioned their own interpretation. I could go on in my criticisms of the paper but will stop at this point to say that I do not believe the paper is suitable for publication within the journal and I cannot see how the authors can revisit the work to make it acceptable. Such a paper may have greater value to a Social Sciences journal rather than a journal directed towards clinicians, particularly involved with the management of headache.
--	--

VERSION 2 – AUTHOR RESPONSE

Reviewer 1 We thank you very much for your intuitive and kind remarks about our paper. In the table below we have responded to your specific queries and comments.	
Reviewer 1 comments	Response
P.; line 6: I was not able to identify any other similar review	Thank you for your confirmation of this
p.3; line 15: Description of patients, or lack of, may simply be a reflection of the original purpose and intended audience of the included papers.	Thank you for this observation we agree.
p.5, line 8: acronym MOH not previously introduced	We have amended this by adding text
p.5, line 28: January 1988 is a reasonable start date and clear rationale given.	We agree
p.6, line 9: missing word 'included' after 'were'?	We have reworded the text for clarity

p.6, line 35 onwards: clearly sets out the issues in selecting studies and the pragmatic approach adopted to selecting studies for inclusion.	We agree – thank you
p.7, line 29 onwards: good accessible description of synthesis approach adopted that clearly identifies methodological issues for the reader.	We agree – thank you
p.10, first paragraph. Did the team consider whether there were issues related to combining data from focus groups with data from individual interviews?	You raise an interesting point. As we were primarily dealing with patient data and generated themes we did not feel that we needed to make this demarcation.
p.17, Discussion identifies and addresses limitations of the study.	We agree – thank you
Figure 2 – development of third order codes – will this be in colour to aid the reader?	An excellent idea, thank you. We have updated this; we hope this is acceptable and helps.
Some page numbers for supporting quotes in appendix are missing.	We have checked and amended this
Reviewer 2 The team would like to thank the reviewer for taking the time to review our paper and providing us with such frank feedback.	
Response We have considered the issues you raised about the use of diagnostic groups but feel we have covered this issue sufficiently in the paper. We note the concerns about the methodology and its appropriateness in medical literature. Meta-ethnography is a well-established methodology. It has a history going back to the 1980s when the first major text on this methodology was published in the education field (Noblit & Hare 1988). Since this date it has developed considerably and is now commonly used in healthcare reviews and there is a growing number of published qualitative meta-ethnography papers related to health. To illustrate this we provide a number of examples at the end of the document. We feel that papers like these are of interest to a clinical as well as research audiences. Although we only found four papers this does not detract from the method, indeed it is a telling result. The aim of the study was to find out what people with chronic headache experience. From the literature it seems they have rarely been asked. The lack of findings are perhaps as important as the findings in this piece of work. It is of interest that four such different studies produced overlapping themes and could be synthesized. RE comment on English language papers: Thank you for your comment. We did include studies in French, German and Spanish to be as inclusive as possible with the access we had to researchers who spoke these languages fluently. In the searches no other languages came up therefore no foreign language studies were excluded. We have added text in the results section to explain this. RE comment about headache type: A reductionist approach of isolating individual headache types and applying today's diagnostic criteria would mean we would have no information rather than some. We suggest that the experience of living with chronic headaches is not substantially changed by the underpinning diagnosis. Whilst accepting the chicken and egg issue we need to report what the studies found and as that was not their take on the data we cannot add this to the analysis.	

The terms "chronic headache" and "chronic daily headache" are not synonymous and to accept them as such is to defeat the purpose behind the studies.

We have included "chronic headache" and "chronic daily headache" in this review as we aimed to be as inclusive as possible rather than to suggest these were diagnostically synonymous terms. We have changed the text accordingly.

Examples below.

Campbell, Rona, et al. "Evaluating meta-ethnography: a synthesis of qualitative research on lay experiences of diabetes and diabetes care." Social science & medicine 56.4 (2003): 671-684.

Elmir, Rakime, et al. "Women's perceptions and experiences of a traumatic birth: a meta-ethnography." Journal of advanced nursing 66.10 (2010): 2142-2153.

Cosco, Theodore D., et al. "Lay perspectives of successful ageing: a systematic review and meta-ethnography." BMJ open 3.6 (2013): e002710.

Sinnott, Carol, et al. "GPs' perspectives on the management of patients with multimorbidity: systematic review and synthesis of qualitative research." BMJ open 3.9 (2013): e003610.